# Potential risk factors for cardiovascular diseases and associated sociodemographic characteristics: A cross-sectional evaluation of a large cohort of women living with HIV in north-central Nigeria

**Olufemi Ajumobi**[1], **Ijeoma Uchenna Itanyi**[2,3], **Amaka Grace Ogidi**[3], **Samantha A. Slinkard**[4], **Echezona Edozie Ezeanolue**[3]*

**1** School of Public Health, University of Nevada, Reno, Nevada, United States of America, **2** Department of Community Medicine, College of Medicine, University of Nigeria Nsukka, Enugu, Nigeria, **3** Center for Translation and Implementation Research, University of Nigeria Nsukka, Enugu, Nigeria, **4** South Carolina Center for Rural and Primary Healthcare, Columbia, South Carolina, United States of America

* eezeanolue@gmail.com

**Data Availability Statement:** Our data involves human research participants of a vulnerable

## Abstract

Males have a higher prevalence of cardiovascular (CVD) risk factors such as alcohol use, hypercholesterolemia, hypertension, obesity, and smoking based on limited data available from two tertiary health centers in Nigeria. Increasing age and lower educational level influence smoking among the same population in northeastern and northwestern Nigeria. Specifically in women living with HIV (WLHIV), the association between demographic characteristics and CVD risk factors has not been described. In a multi-center cross-sectional study, we documented the association of sociodemographic characteristics with potential CVD risk factors among a large cohort of WLHIV attending five treatment sites in north-central Nigeria. This was a cross-sectional study among 5430 women of reproductive age who received antiretrovirals at five selected treatment sites in Benue State, Nigeria. We performed multivariable regression of sociodemographic characteristics on potential cardiovascular risk factors, namely, smoking, alcohol consumption, and contraceptive use. We found participants' mean age was 33.2 (standard deviation: 6.1) years. Prevalence of smoking, alcohol consumption, and contraceptive use were 0.6%, 11%, and 7% respectively. Older WLHIV ($\geq$ 40 years) had a negative association with contraceptive use (aOR: 0.58, 95%CI: 0.42–0.81). Being educated WLHIV had a positive association with contraceptive use (aOR: 1.34, 95%CI: 1.02–1.76) and a negative association with tobacco smoking (aOR: 0.37, 95%CI: 0.16–0.83). Being a farmer had a negative association with alcohol consumption (aOR: 0.43, 95%CI: 0.35–0.52) and contraceptive use (aOR: 0.61, 95%CI: 0.48–0.76). Compared to being married, being in a single relationship had positive association with alcohol consumption (aOR: 1.30, 95%CI: 1.08–1.56) while parenting was associated with 165% higher odds of contraceptive use (aOR: 2.65, 95%CI: 1.73–4.06). In conclusion, the low prevalence of smoking exists among women living with HIV on antiretroviral treatment. Older age, farming and being married are potential deterrents to lifestyle risk factors for

population. These are women in rural areas of Nigeria living with HIV and or HBV. They are prone to stigmatization in spite of data de-identification. As a result of this, there are ethical restrictions on making the data publicly available. Data access requests can be made to our institutional body. Name: Centre for Translation and Implementation Research, College of Medicine, University of Nigeria, Nsukka, Enugu State, Nigeria (UNN). Email: ctair@unn.edu.ng.

**Funding:** EEE received funding support from Fogarty International Center of the US National Institutes of Health (NIH) (grant no. R21TW010252) and the Eunice Kennedy Shriver National Institute of Child Health & Human Development of the NIH (grant numbers - R01HD087994 & R01HD087994-S1). The content is solely the responsibility of the authors and does not necessarily represent the official views of the NIH. The funders had no role in study design, data collection and analysis, decision to publish, or preparation of the manuscript.

**Competing interests:** The authors have read the journal's policy and have the following competing interests: The smart card, 'MyHealthCard' (https://myhealthcard.io/) has been licensed to Lion Health (https://lionhealth.io/), and consistent with policy, a portion of royalty is distributed to EEE. This does not alter our adherence to PLOS policies on sharing data and materials. All other authors have declared that no competing interests exist.

cardiovascular diseases among this population. To improve HIV-related treatment efforts and outcomes, implementing interventions aimed at lifestyle behavioral modification among this population has the potential to reduce cardiovascular disease risks.

## Introduction

Cardiovascular diseases have become a major health concern for persons living with HIV (PLHIV) especially because antiretroviral therapy (ART) is more accessible than it was two decades ago and has contributed to living longer with the disease. Among other factors, chronic immune activation and inflammation associated with HIV contribute to cardiovascular atherosclerosis [1]. Globally, the burden of cardiovascular disease has tripled over the past two decades [2]. In Nigeria, common cardiovascular (CVD) risk factors are hypertension, dyslipidemia, and low physical activity [3, 4].

Among both antiretroviral (ARV) naïve and PLHIV receiving routine highly active antiretroviral treatment (HAART), males have a higher prevalence of CVD risk factors such as alcohol use, diabetes mellitus, dyslipidemia, hypercholesterolemia, hypertension, obesity, and smoking [3–7]. These data were limited to a tertiary health center each in a state in the northern and southern part of Nigeria and these were not representative of those states. Increasing age and lower educational level are risk factors for smoking among mainly male PLHIV surveyed in single health facilities in northeastern and north-western Nigeria [7, 8]. But the influence of these factors on smoking and other potential CVD risk factors may differ among female PLHIV in other zones of the country. Additionally, the influence of occupation, marital status, and parenthood on the development of potential CVD risk factors among PLHIV in Nigeria has not been established. Our multi-center study documented the prevalence of smoking, alcohol consumption, and contraceptive use among a large cohort of women living with HIV (WLHIV) in north-central Nigeria and the association with sociodemographic characteristics.

## Methods

### Study context, design, and participants

The current study leveraged the scale-up of an integrated mhealth intervention for a cohort of HIV-infected women of reproductive age receiving care at five comprehensive HIV treatment centers in Benue State, Nigeria. The development of the integrated mHealth platform was described in a previous study [9]. Summarily, the mHealth platform could store encrypted patient information on a patient-held smartcard and smartphone mobile application, which is linked with a secure web-based engagement management database [9, 10]. Patient records can be viewed on the mobile app without an internet connection, and this aids in healthcare decision-making at the point of care in low-resource settings [10]. Benue State has an estimated total population of 5,138,531, of whom 49.6% are female. Most of the population resides in rural areas and the majority, about 75%, are farmers [11]. About 70% of the female population has less than a secondary school education with a literacy rate of 52.8% [12].

Each study site was selected based on (1) designation as a a comprehensive HIV treatment facility, (2) receipt of funding support from US President's Emergency Plan for AIDS Relief (PEPFAR) through Caritas Nigeria, (3) records of a high volume of HIV-infected women ($\geq$ 2000 women on treatment), and (4) provision of free HIV testing services, antiretroviral

therapy for both adults and children and services for prevention of mother-to-child transmission (MTCT). There were 21 comprehensive HIV treatment facilities in Benue State, seven of which had at least 2000 women on treatment. Administrative leads of five out of these seven facilities gave approval for participation in the study and those facilities were selected. All HIV-infected women were eligible to participate in this study if they were of reproductive age (18–45 years) and received ART from any of the five selected study sites.

### Data collection

Trained health workers offered pre-printed mhealth smartcards with unique patient identifiers to all eligible HIV-infected women as they presented for their routine pre-scheduled clinic appointments. The study's purpose and procedures were explained to potential participants and those who gave written informed consent were chosen to participate in the study.

Trained research assistants administered pretested semi-structured questionnaires to the study participants and collected information on the sociodemographic and clinical characteristics, and lifestyle habits. Sociodemographic characteristics included age, marital status, the highest level of education, occupation, and the number of living children. Clinical characteristics included the current ART regimen, use of contraceptives, whether the participant was currently pregnant, and whether antenatal care was received for the current pregnancy. Participants were also asked about their tobacco smoking and alcohol habits. This study was carried out between June and December 2017.

### Data analysis

Data on 5430 WLHIV were analyzed using SAS version 9.4. We described participants' sociodemographic characteristics and the prevalence of clinical characteristics and potential cardiovascular risk factors. Potential risk factors for CVD are age-related and a cut-off of 40 years has been used in prior studies cut-off [6, 7]. Other than age, the association between independent variables such as education (educated/none), occupation (farmers/non-farmers), and having children (≥1/none) [13], and potential cardiovascular risk factors, namely, smoking, alcohol consumption, and contraceptive use (coded: Yes/No) were examined at bivariate and multivariable logistic regression analyses. Results were presented using odds ratio (OR) and adjusted OR. We assessed the fitness of each of our models using the global null hypothesis test leveraging the approach in previous studies and there were no issues of multicollinearity [14, 15]. Statistical significance was considered at a p-value of less than 0.05.

### Ethical considerations

Ethical approval for this study was obtained from the Health Research Ethics Committee of the University of Nigeria Teaching Hospital (NHREC/05/01/2008B-FWA00002458-1RB00002323). Approval was obtained from the health administrators of all the study sites and written informed consent was obtained from participants.

## Results

The mean age of participants was 33.2 (standard deviation [SD]: 6.1) years. Most were married (66.9%, n = 3632), completed high school education (24.3%, n = 1317), were married (6.9%, n = 3632), were farmers (68.4%, n = 3712), and had an average of 3 children (SD: 2), see Table 1.

**Table 1. Sociodemographic characteristics of HIV-infected women in Benue, Nigeria (N = 5430).**

| Characteristics | n (%) |
|---|---|
| **Age** | |
| 18–20 | 61 (1.1%) |
| 21–25 | 545 (10.0) |
| 26–30 | 1247 (23.0) |
| 31–35 | 1614 (29.7) |
| 36–40 | 1175 (21.6) |
| 41–45 | 788 (14.5) |
| **Marital status** | |
| Single | 614 (11.3) |
| Married | 3632 (66.9) |
| Divorced | 430 (7.9) |
| Widowed | 754 (13.9) |
| **Number of children** | |
| None | 683 (12.6) |
| 1–2 | 1851 (34.1) |
| 3–4 | 1696 (31.2) |
| 5+ | 1200 (22.1) |
| **Highest educational level attained** | |
| None | 1532 (28.2) |
| Completed primary | 1337 (24.6) |
| Completed junior secondary | 869 (16.0) |
| Completed senior secondary | 1018 (18.8) |
| Completed post-secondary | 299 (5.5) |
| Some post-secondary | 375 (6.9) |
| **Occupation** | |
| Civil Servant | 333 (6.1) |
| Farmer | 3712 (68.4) |
| Trader | 998 (18.4) |
| Applicant | 146 (2.7) |
| Other* | 241 (4.4) |
| **Alcohol consumption** | |
| Yes | 590 (10.9) |
| No | 4840 (89.1) |
| **Cigarette smoking** | |
| Yes | 30 (0.6) |
| No | 5400 (99.4) |
| **Secondhand smoke (n = 4169)** | |
| daily | 665 (16.0) |
| weekly | 19 (0.5) |
| monthly | 6 (0.1) |
| less than once a month | 2 (0.1) |
| never | 3477 (83.4) |

Other

*: fashion designers, hairdressers, health workers, housewives, midwives, public servants, students, and tailors

## Cardiovascular risk factors

Regarding social/lifestyle characteristics, nearly all participants (99.4%, n = 5400) were non-smokers. The prevalence of cigarette smoking and alcohol was 0.6% and 10.9% respectively. About 16.7% (n = 692) of the participants experienced secondhand/passive smoking (Table 1). Overall, 5360 participants had one out of three potential cardiovascular risk factors: cigarette smoking, alcohol consumption, and contraceptive use, 70 had any two of three factors and none had the three risk factors.

## Clinical characteristics

Overall, 7% (379/5430) were using contraceptives, 4.9% (264/5430) were pregnant and 66.3% (175/264) were attending antenatal care. Of the 5430 WLHIV, 5387 (99.2%) were currently on ART regimens. Of these, 5084 (94.4%) were still receiving first-line antiretroviral therapy, with two-thirds receiving tenofovir + Lamivudine + Efavirenz combination (4%) and one-third receiving Zidovudine + Lamivudine + Nevirapine combination (Table 2).

## Smoking, alcohol consumption, and contraceptives use and associated factors

In multivariable analysis after controlling for other variables, being educated had 63% lower odds of smoking. Being a farmer had 57% lower odds of alcohol consumption while a single relationship increased the odds of alcohol consumption by 30%.

Those who were 40 years and above had 42% lower odds of contraceptive use unlike the educated who had 34% higher odds of contraceptive use. Being a farmer had 39% lower odds of contraceptive use while parenting had 165% higher odds of contraceptive use (Table 3).

**Table 2. Clinical characteristics of HIV-infected women in Benue, Nigeria (N = 5430).**

| Characteristics | n (%) |
|---|---|
| **Use contraceptives** | |
| No | 5051 (93.0) |
| Yes | 379 (7.0) |
| **Currently pregnant** | |
| Yes | 264 (4.9) |
| No | 5166 (95.1) |
| **Attending ANC (n = 264)** | 175 (66.3%) |
| **Category of ART regimen (n = 5387)** | |
| 1st line | 5084 (94.4) |
| 2$^{nd}$ line | 303 (5.6) |
| **Current ART regimen (n = 5387)** | |
| TDF/3TC/EFV | 3432 (63.7) |
| AZT/3TC/NVP | 1648 (30.6) |
| TDF/3TC/LOP/r | 142 (2.6) |
| AZT/3TC/LOP/r | 134 (2.5) |
| TDF/3TC/ATV/r | 15 (0.3) |
| AZT/3TC/ATV/r | 5 (0.1) |
| ABC/3TC/LOP/r | 6 (0.1) |
| ABC/3TC/EFV | 4 (0.1) |
| ABC/3TC/ATV/r | 1 (0.0) * |

*Rounding-off value

**Table 3. Sociodemographic characteristics associated with smoking, alcohol consumption, and contraceptive use among HIV-infected women in Benue, Nigeria.**

| Characteristics | | Smoking | | Alcohol | | Contraceptive | |
|---|---|---|---|---|---|---|---|
| | | OR (95%CI) | aOR (95%CI) | OR (95%CI) | aOR (95%CI) | OR (95%CI) | aOR (95%CI) |
| Age | ≥ 40 vs 18–39 | 0.68 (0.24–1.97) | 0.64 (0.22–1.87) | 0.95 (0.76–1.19) | 1.02 (0.81–1.28) | **0.57 (0.41–0.79)**\* | **0.58 (0.42–0.81)**\* |
| Education | Educated vs none | 0.51 (0.25–1.06) | **0.37 (0.16–0.83)**\* | **1.37 (1.12–1.67)**\* | 1.00 (0.80–1.24) | **1.59 (1.23–2.05)**\* | **1.34 (1.02–1.76)**\* |
| Occupation | Farmers vs non-farmers | 0.70 (0.33–1.44) | 0.52 (0.23–1.18) | **0.42 (0.36–0.50)**\* | **0.43 (0.35–0.52)**\* | **0.60 (0.48–0.74)**\* | **0.61 (0.48–0.76)**\* |
| Marital status | Single relationship vs married | 1.17 (0.56–2.47) | 1.09 (0.51–2.36) | **1.40 (1.17–1.67)**\* | **1.30 (1.08–1.56)**\* | 0.92 (0.73–1.15) | 1.00 (0.79–1.26) |
| Parenthood | Parent vs childless adult | 0.72 (0.27–1.88) | 0.77 (0.28–2.12) | 0.85 (0.67–1.09) | 1.15 (0.88–1.49) | **2.12 (1.40–3.20)**\* | **2.65 (1.73–4.06)**\* |

\*Statistically significant at p <0.05

## Discussion

Our data revealed the prevalence of potential CVD risk factors in decreasing order of frequency: alcohol consumption, contraceptive use, passive smoking, and cigarette smoking. Education and having at least a child were independent risk factors for contraceptive use but older age and being a farmer were protective. Unlike single relationship status, engagement in farming was protective against alcohol consumption. Being educated was a disincentive to cigarette smoking. The demographic factors were associated with the outcomes, namely, contraceptive use, alcohol consumption, and smoking, but not causally related.

The low prevalence of current smoking (0.6%) reported in this study is similar to that reported among WLHIV in Ogbomosho, South-West Nigeria, and half of that reported in northern Nigeria [3]. However, this is the least prevalent potential CVD risk factor among female PLHIV in our study compared to a range of zero prevalence found in Osogbo, South-West Nigeria and Kano North-West Nigeria, 2% in Ghana (Western-Africa), 6% in rural Uganda (Eastern-Africa) and 13% in Klerksdorp, South Africa [3, 8, 16–21]. Appiah and colleagues reported a pooled current smoking prevalence of 1.3% in 28 low-and-middle-income countries excluding Nigeria, South Africa, and Ghana [16]. The low prevalence of smoking among women might be because generally, women in sub-Saharan Africa are less likely to smoke and especially WLHIV [19, 22].

Alcohol consumption is a public concern among WLHIV because of the potential for disease progression [23, 24]. The 11% prevalence of alcohol consumption is lower compared to that reported in South Africa (15%) and globally (female: pooled [13.4%]) but higher than 1.1% reported among WLHIV attending the Ladoke Akintola University of Technology Teaching Hospital in Ogbomosho, a sub-urban HIV care center in South-West Nigeria [3, 21, 25]. The relatively high prevalence of alcohol consumption is concerning. Alcohol use may impair judgment, reduce risk perception, cause disinhibition, and increase the likelihood of unprotected sexual behavior [26, 27]. Also, alcohol use is associated with the reduced efficacy of ARV drugs and medication non-compliance, and therefore, educating PLHIV on these potential consequences is critical [5, 28].

The low prevalence of contraceptive use among our study population would have been a concern for MTCT in our study population but for the fact that nearly all were on HAART. Higher use of intrauterine devices (26.7%) and male condoms (29.1%) have been reported elsewhere [29, 30].

Benue is an agrarian state known as the food basket of the nation [31]. The observed protective effect of farming against alcohol consumption in our study could be because farmers are usually too busy with planting, harvesting, and marketing their products for economic gain. Alcohol consumption is commoner in the evenings by which time farmers are tired from the day's work on the farm. Additionally, they may not want to risk the progression of disease

with alcohol use [27]. However, farming did not foster contraceptive use in our study population. Contraception discourages childbearing and farming, being labor-intensive and non-mechanized, farmers tend to have a large family size. In our study, participants had an average of three children.

Age is a determinant of CVD risk [4, 6]. The decreased likelihood of contraceptive use with older age contrasts with the finding in non-HIV women of reproductive age [32]. However, it is unclear why increasing age is associated with lower odds of contraceptive use.

The single relationship remained a positive predictor of alcohol consumption. In a single relationship, alcohol use may help fill the void of stable companionship which could easily be met with cohabitation in the Western world but is strictly forbidden in the African setting based on sociocultural and religious norms [33, 34] In the African sociocultural landscape, a marital relationship provides social support which is often necessary for a PLHIV [35].

Being a parent had almost three times higher odds of contraceptive use compared to being a childless woman living with HIV. In African society, bringing forth new offspring where zero mother-to-child transmission of HIV cannot be guaranteed is undesirable. Thus, child spacing with contraceptive use is a rational decision to prevent unintended pregnancies and new HIV infections [36]. While elimination remains a target for sub-Saharan Africa, it remains a mirage until optimal MTCT services and sustained access to ARV are guaranteed. Currently, in Nigeria, access to ART is largely donor-driven [37]. Thus, in our study, that being educated fostered contraceptive use, was not a surprise [37].

In our study, education was protective against smoking unlike in prior studies in northern Nigeria which revealed the more educated you were, the higher your odds of smoking [8]. Higher educational attainment and literacy rates are twice as high in Benue State (middle-belt region of Nigeria) compared to other states in northern Nigeria [38]. These correlate with health education and might be responsible for the very low prevalence of cigarette smoking because of the prevailing belief that smokers are liable to die young.

## Limitations of the study

This study had limitations. The degree of alcohol consumption (drinks/week) and cigarette smoking (pack years) was not quantified, and the type of contraceptive use was not specified. The CD4 counts and viral load for the study population were not collected, which could have allowed the possibility of ascertaining the relationship between the prevalent potential CVD risk factors and HIV progression in an entire female HIV population. This study was carried out in facilities where we received administrative approval and we therefore acknowledge the potential for selection bias. We did not study other risk factors for CVD such as hypertension, obesity, and hypercholesterolemia. Moreover, the study had some strengths. Most similar studies in Africa were conducted in single HIV treatment centers [3, 17, 21]. In addition, there is a paucity of studies that have examined the association between demographic characteristics and potential CVD risk factors such as smoking, alcohol, and contraceptive use among WLHIV. Data on these factors are not routinely collected for PLHIV [4]. Our study was a multi-center study with a large dataset of over 5000 participants. In comparison with single facility-based studies, this study provided more robust information and precision of estimates and the findings are generalizable to similar settings.

## Conclusions

A low prevalence of smoking exists among women living with HIV on antiretroviral treatment. Older age, farming and being married are potential deterrents to lifestyle risk factors for cardiovascular diseases among this population. The goal is to improve HIV-related treatment

efforts and outcomes for this population. Implementing interventions that could foster lifestyle behavioral modification among women living with HIV on antiretroviral treatment has the potential to reduce cardiovascular disease risks.

## Acknowledgments

The authors acknowledge the support from the heads and staff of the participating health facilities, the patients, the staff of Caritas Nigeria, and the staff of the Center for Translation and Implementation Research (CTAIR) of the University of Nigeria, Nsukka, Enugu.

## Author Contributions

**Conceptualization:** Ijeoma Uchenna Itanyi, Echezona Edozie Ezeanolue.

**Data curation:** Amaka Grace Ogidi, Echezona Edozie Ezeanolue.

**Formal analysis:** Olufemi Ajumobi, Samantha A. Slinkard.

**Funding acquisition:** Echezona Edozie Ezeanolue.

**Methodology:** Olufemi Ajumobi, Ijeoma Uchenna Itanyi, Echezona Edozie Ezeanolue.

**Software:** Olufemi Ajumobi.

**Supervision:** Echezona Edozie Ezeanolue.

**Visualization:** Olufemi Ajumobi, Echezona Edozie Ezeanolue.

**Writing – original draft:** Olufemi Ajumobi, Ijeoma Uchenna Itanyi.

**Writing – review & editing:** Olufemi Ajumobi, Ijeoma Uchenna Itanyi, Amaka Grace Ogidi, Samantha A. Slinkard, Echezona Edozie Ezeanolue.

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
