## [Decision Letter · Decision Letter 0]

20 Dec 2022

PGPH-D-22-01427

Risk factors for Cardiovascular Diseases and the Potential Influence of Sociodemographic characteristics: a cross-sectional evaluation of a large cohort of Women Living with HIV in north-central Nigeria

Dear Dr. Ezeanolue,

Thank you for submitting your manuscript to PLOS Global Public Health. After careful consideration, we feel that it has merit but does not fully meet PLOS Global Public Health’s publication criteria as it currently stands. Therefore, we invite you to submit a revised version of the manuscript that addresses the points raised during the review process.

We look forward to receiving your revised manuscript.

Kind regards,

Abraham D. Flaxman, Ph.D.

Academic Editor

Journal Requirements:

1. Please send a completed 'Competing Interests' statement, including any COIs declared by your co-authors. If you have no competing interests to declare, please state "The authors have declared that no competing interests exist". Otherwise please declare all competing interests beginning with the statement "I have read the journal's policy and the authors of this manuscript have the following competing interests:"

Additional Editor Comments (if provided):

Reviewers' comments:

Reviewer's Responses to Questions

**Comments to the Author**

1. Does this manuscript meet PLOS Global Public Health’s publication criteria? Is the manuscript technically sound, and do the data support the conclusions? The manuscript must describe methodologically and ethically rigorous research with conclusions that are appropriately drawn based on the data presented.

Reviewer #1: Yes

2. Has the statistical analysis been performed appropriately and rigorously?

Reviewer #1: Yes

3. Have the authors made all data underlying the findings in their manuscript fully available (please refer to the Data Availability Statement at the start of the manuscript PDF file)?

Reviewer #1: Yes

4. Is the manuscript presented in an intelligible fashion and written in standard English?

Reviewer #1: No

5. Review Comments to the Author

Reviewer #1: Author review

Title: Risk factors for Cardiovascular Diseases and the Potential Influence of Sociodemographic characteristics: a cross-sectional evaluation of a large cohort of Women Living with HIV in north-central Nigeria

Suggestions and comments

Abstract: -

“Background: Age and education are known cardiovascular (CVD) risk factors in persons living with HIV (PLHIV).” This statement should be rewritten b/c when you say education is a known CVD risk factors it is not clear. Is it lack of information about CVDs? or having higher educational level, other?

“We performed regression analyses of the influence of sociodemographic characteristics on potential cardiovascular risk factors, namely, smoking, alcohol consumption, contraceptive use.” Why you didn’t consider other CVDs risk factors like obesity (high cholesterol level/ higher BMI, sedentary life/lack of regular physical activity, hypertension and …………...? Suggestion: if you have no reason/s justification add it as limitation.

On conclusion “Targeting educated WLHIV who are smokers and alcohol users in single relationships has the potential to foster the maintenance of viral suppression, reduce CVD and improve treatment outcomes in WLHIV.” Is scientifically acceptable to keep smokers and alcoholics in single relationships instead of looking for another solution? Suggestion: look up any written guidelines/ articles and rewrite it.

Introduction: -

Paragraph 2 and 3 lack coherence and need edition for English language.

Methods: -

“This study was carried out between June and December 2017.” Write the specific date of data collection.

“Independent variables included age (18-39/≥40years), education (educated/none), occupation (farmers/non-farmers), and having children (≥1/none).” What is your evidence to categorize this predictor variables? cite it.

Results: -

Write table footnote for those “other” in the result table.

Link result with table (cite result).

Clear explain which variables are analyzed by multivariate and what is the criteria to be analyzed by it after binary regression analysis i.e either indicate cut off P- value, or…...

The authors wrote, inferential statistics was obtained by computing bivariate /multivariate regression. Please write explicitly what type of regression (linear vs binary logistic) and model fitness/ procedures followed while you perform analysis.

Conclusion: -

Conclusion should be written based on the finding.

6. PLOS authors have the option to publish the peer review history of their article (what does this mean?). If published, this will include your full peer review and any attached files.

**Do you want your identity to be public for this peer review?** For information about this choice, including consent withdrawal, please see our Privacy Policy.

Reviewer #1: No

---

## [Decision Letter · Decision Letter 1]

7 May 2023

PGPH-D-22-01427R1

Risk factors for Cardiovascular Diseases and the Potential Influence of Sociodemographic characteristics: a cross-sectional evaluation of a large cohort of Women Living with HIV in north-central Nigeria

Dear Dr. Ezeanolue,

Thank you for submitting your manuscript to PLOS Global Public Health. After careful consideration, we feel that it has merit but does not fully meet PLOS Global Public Health’s publication criteria as it currently stands. Therefore, we invite you to submit a revised version of the manuscript that addresses the points raised during the review process.

We look forward to receiving your revised manuscript.

Kind regards,

Jianhong Zhou

Staff Editor

Journal Requirements:

1. Please send a completed 'Competing Interests' statement, including any COIs declared by your co-authors. If you have no competing interests to declare, please state "The authors have declared that no competing interests exist". Otherwise please declare all competing interests beginning with the statement "I have read the journal's policy and the authors of this manuscript have the following competing interests:"

Additional Staff Editor Comments (if provided): Please note we invited a new reviewer for this revision in addition to the previous reviewer. The new reviewer raised a few concerns which should be addressed.

Reviewers' comments:

Reviewer's Responses to Questions

**Comments to the Author**

1. If the authors have adequately addressed your comments raised in a previous round of review and you feel that this manuscript is now acceptable for publication, you may indicate that here to bypass the “Comments to the Author” section, enter your conflict of interest statement in the “Confidential to Editor” section, and submit your "Accept" recommendation.

Reviewer #1: All comments have been addressed

Reviewer #2: (No Response)

2. Does this manuscript meet PLOS Global Public Health’s publication criteria? Is the manuscript technically sound, and do the data support the conclusions? The manuscript must describe methodologically and ethically rigorous research with conclusions that are appropriately drawn based on the data presented.

Reviewer #1: Yes

Reviewer #2: (No Response)

3. Has the statistical analysis been performed appropriately and rigorously?

Reviewer #1: Yes

Reviewer #2: (No Response)

4. Have the authors made all data underlying the findings in their manuscript fully available (please refer to the Data Availability Statement at the start of the manuscript PDF file)?

Reviewer #1: Yes

Reviewer #2: (No Response)

5. Is the manuscript presented in an intelligible fashion and written in standard English?

Reviewer #1: Yes

Reviewer #2: (No Response)

6. Review Comments to the Author

Reviewer #1: (No Response)

Reviewer #2: (No Response)

7. PLOS authors have the option to publish the peer review history of their article (what does this mean?). If published, this will include your full peer review and any attached files.

**Do you want your identity to be public for this peer review?** For information about this choice, including consent withdrawal, please see our Privacy Policy.

Reviewer #1: No

Reviewer #2: No

---

## [Decision Letter · Decision Letter 2]

17 Jul 2023

PGPH-D-22-01427R2

Potential Risk factors for Cardiovascular Diseases and associated Sociodemographic characteristics: a cross-sectional evaluation of a large cohort of Women Living with HIV in north-central Nigeria

Dear Dr. Ezeanolue,

Thank you for submitting your manuscript to PLOS Global Public Health. After careful consideration, we feel that it has merit but does not fully meet PLOS Global Public Health’s publication criteria as it currently stands. Therefore, we invite you to submit a revised version of the manuscript that addresses the points raised during the review process.

We look forward to receiving your revised manuscript.

Kind regards,

Miquel Vall-llosera Camps

Staff Editor

Journal Requirements:

Reviewers' comments:

Reviewer's Responses to Questions

**Comments to the Author**

1. If the authors have adequately addressed your comments raised in a previous round of review and you feel that this manuscript is now acceptable for publication, you may indicate that here to bypass the “Comments to the Author” section, enter your conflict of interest statement in the “Confidential to Editor” section, and submit your "Accept" recommendation.

Reviewer #2: All comments have been addressed

2. Does this manuscript meet PLOS Global Public Health’s publication criteria? Is the manuscript technically sound, and do the data support the conclusions? The manuscript must describe methodologically and ethically rigorous research with conclusions that are appropriately drawn based on the data presented.

Reviewer #2: Partly

3. Has the statistical analysis been performed appropriately and rigorously?

Reviewer #2: Yes

4. Have the authors made all data underlying the findings in their manuscript fully available (please refer to the Data Availability Statement at the start of the manuscript PDF file)?

Reviewer #2: Yes

5. Is the manuscript presented in an intelligible fashion and written in standard English?

Reviewer #2: Yes

6. Review Comments to the Author

Reviewer #2: Dear Editor,

I have reviewed the revised manuscript entitled "Potential Risk factors for cardiovascular diseases and associated Sociodemographic characteristics: a cross-sectional evaluation of a large cohort of Women Living with HIV in north-central Nigeria".

The paper has undergone revisions to address the concerns raised during the initial review. The authors have been receptive to feedback and made some efforts to improve the manuscript.

In the revised paper, the authors have refrained from limiting their conclusion to smokers and alcohol users and emphasized the need for lifestyle behavioral modifications among women living with HIV on antiretroviral treatment.

The authors clarified that they meant “potential cardiovascular risk factors” and corrected this in the title and the manuscript.

The revised paper now includes confidence intervals for the estimates which makes it more informative.

The authors have added information about the chronic immune activation and inflammation associated with HIV and antiretroviral treatment, contributing to cardiovascular atherosclerosis.

The authors explained that age was dichotomized to enable comparability with earlier studies and referenced prior studies.

The authors included an explanation for the global null hypothesis test and how it is used to assess the fit of models, along with citations.

The authors updated the limitation section to include potential selection bias and the inability to study other major risk factors for CVD such as hypertension, obesity, and hypercholesterolemia.

Overall, the authors have made some efforts to address the issues raised in the initial review. However, I still have some concerns about the objective of the study and the connection between the findings and conclusion. For example, how can the following findings be useful? “Being a farmer negatively predicted alcohol consumption”, “Compared to being in a single relationship, being married positively predicted alcohol consumption”, “Education and having at least a child were independent risk factors for contraceptive use but older age and being a farmer were protective”. This seems to imply that we should encourage people to be a farmer and not get married. Furthermore, how could the conclusion reflect the findings more concretely?

I look forward to seeing the further revisions.

7. PLOS authors have the option to publish the peer review history of their article (what does this mean?). If published, this will include your full peer review and any attached files.

**Do you want your identity to be public for this peer review?** For information about this choice, including consent withdrawal, please see our Privacy Policy.

Reviewer #2: No

---

## [Decision Letter · Decision Letter 3]

20 Sep 2023

PGPH-D-22-01427R3

Potential Risk factors for Cardiovascular Diseases and associated Sociodemographic characteristics: a cross-sectional evaluation of a large cohort of Women Living with HIV in north-central Nigeria

Dear Dr. Ezeanolue,

Thank you for submitting your manuscript to PLOS Global Public Health. After careful consideration, we feel that it has merit but does not fully meet PLOS Global Public Health’s publication criteria as it currently stands. Therefore, we invite you to submit a revised version of the manuscript that addresses the points raised during the review process.

Reviewer 2 still has some issues that need addressing - please see their comments below.

Could you please revise the manuscript to carefully address the concerns raised?

We look forward to receiving your revised manuscript.

Kind regards,

Steve Zimmerman, PhD

PLOS Staff Editor

Journal Requirements:

2. Please send a completed 'Competing Interests' statement, including any COIs declared by your co-authors. If you have no competing interests to declare, please state "The authors have declared that no competing interests exist". Otherwise please declare all competing interests beginning with twhe statement "I have read the journal's policy and the authors of this manuscript have the following competing interests:"

Additional Editor Comments (if provided):

Reviewers' comments:

Reviewer's Responses to Questions

**Comments to the Author**

1. If the authors have adequately addressed your comments raised in a previous round of review and you feel that this manuscript is now acceptable for publication, you may indicate that here to bypass the “Comments to the Author” section, enter your conflict of interest statement in the “Confidential to Editor” section, and submit your "Accept" recommendation.

Reviewer #2: (No Response)

2. Does this manuscript meet PLOS Global Public Health’s publication criteria? Is the manuscript technically sound, and do the data support the conclusions? The manuscript must describe methodologically and ethically rigorous research with conclusions that are appropriately drawn based on the data presented.

Reviewer #2: (No Response)

3. Has the statistical analysis been performed appropriately and rigorously?

Reviewer #2: (No Response)

4. Have the authors made all data underlying the findings in their manuscript fully available (please refer to the Data Availability Statement at the start of the manuscript PDF file)?

Reviewer #2: (No Response)

5. Is the manuscript presented in an intelligible fashion and written in standard English?

Reviewer #2: (No Response)

6. Review Comments to the Author

Reviewer #2: Major Issues:

1. Contradictory Results on Alcohol Consumption:

The paper presents contradictory findings regarding the relationship between marital status and alcohol consumption. Line 173 states that being in a marital relationship increases the odds of alcohol consumption by 30%, while line 46 suggests that being single positively predicts alcohol consumption. This inconsistency needs to be addressed and clarified.

2. Unclear Relationship Between Contraceptive Use and Cardiovascular Risk:

The paper briefly mentions that oral contraceptives predispose individuals to cardiovascular diseases (CVD) by causing thrombosis but the study lacks data on the number of subjects who have taken oral contraceptives as opposed to other measures.

3. Data Selection and Potential Bias:

The study includes 5,430 complete-case observations out of a total of 8,825, without providing a rationale for this selection or discussing the potential consequences of dropping the remaining observations. To address this issue, the authors should either justify the exclusion of these observations or consider techniques like data imputation or sensitivity analysis.

Minor Issues:

1. Unrelated Conclusion on Viral Suppression:

The conclusion mentions the potential for lifestyle behavioral modifications to maintain viral suppression, a topic not covered in the study. This statement should either be removed or substantiated with relevant data or discussion.

2. Language Implying Causality:

Although the study admits it cannot establish causal relationships, phrases like "Being a farmer negatively predicted alcohol consumption" imply causality. The language should be revised to reflect the correlational nature of the findings.

3. Unclear Statement on p-value:

The statement "A p-value of less than 0.05 was considered statistically significant, indicating the models were fit" is unclear. The authors should clarify what is meant by "the models were fit" in the context of a p-value less than 0.05.

4. Irrelevant Information on Tobacco Industry:

The statement in line 209, "potentially, they are a target of the Tobacco industry," seems irrelevant to the study's focus and should be removed unless its relevance can be demonstrated.

5. Archaic Language:

The use of archaic words like "thrice" is not recommended in scientific writing. Please consider using more contemporary terms.

Summary:

The manuscript has several major and minor issues that need to be addressed before it can be considered for publication. Clarifying contradictory results, providing a rationale for data selection, and revising language to better align with the study's limitations are essential steps for improvement.

7. PLOS authors have the option to publish the peer review history of their article (what does this mean?). If published, this will include your full peer review and any attached files.

**Do you want your identity to be public for this peer review?** For information about this choice, including consent withdrawal, please see our Privacy Policy.

Reviewer #2: No

---

## [Decision Letter · Decision Letter 4]

6 Nov 2023

Potential Risk factors for Cardiovascular Diseases and associated Sociodemographic characteristics: a cross-sectional evaluation of a large cohort of Women Living with HIV in north-central Nigeria

PGPH-D-22-01427R4

Dear Prof Ezeanolue,

We are pleased to inform you that your manuscript 'Potential Risk factors for Cardiovascular Diseases and associated Sociodemographic characteristics: a cross-sectional evaluation of a large cohort of Women Living with HIV in north-central Nigeria' has been provisionally accepted for publication in PLOS Global Public Health.

Best regards,

Julia Robinson

Executive Editor

Reviewer Comments (if any, and for reference):

Reviewer's Responses to Questions

**Comments to the Author**

1. If the authors have adequately addressed your comments raised in a previous round of review and you feel that this manuscript is now acceptable for publication, you may indicate that here to bypass the “Comments to the Author” section, enter your conflict of interest statement in the “Confidential to Editor” section, and submit your "Accept" recommendation.

Reviewer #2: All comments have been addressed

2. Does this manuscript meet PLOS Global Public Health’s publication criteria? Is the manuscript technically sound, and do the data support the conclusions? The manuscript must describe methodologically and ethically rigorous research with conclusions that are appropriately drawn based on the data presented.

Reviewer #2: Yes

3. Has the statistical analysis been performed appropriately and rigorously?

Reviewer #2: Yes

4. Have the authors made all data underlying the findings in their manuscript fully available (please refer to the Data Availability Statement at the start of the manuscript PDF file)?

Reviewer #2: No

5. Is the manuscript presented in an intelligible fashion and written in standard English?

Reviewer #2: Yes

6. Review Comments to the Author

Reviewer #2: I have had the opportunity to review your manuscript titled "Potential Risk factors for Cardiovascular Diseases and associated Sociodemographic characteristics: a cross-sectional evaluation of a large cohort of Women Living with HIV in north-central Nigeria" through its various stages of revision. Thank you for your efforts in addressing the concerns and suggestions provided in the previous review rounds. I am pleased to see that the paper has improved considerably and now meets the standards of the journal.

7. PLOS authors have the option to publish the peer review history of their article (what does this mean?). If published, this will include your full peer review and any attached files.

**Do you want your identity to be public for this peer review?** For information about this choice, including consent withdrawal, please see our Privacy Policy.

Reviewer #2: No
